# Outcome of Debulking the Mesenteric Mass in Symptomatic Patients with Locally Advanced Small Intestine Neuroendocrine Tumors

**DOI:** 10.3390/cancers17081318

**Published:** 2025-04-14

**Authors:** Detlef K. Bartsch, Norman Krasser-Gercke, Anja Rinke, Andreas Mahnken, Moritz Jesinghaus, Friederike Eilsberger, Elisabeth Maurer

**Affiliations:** 1Department of Visceral-, Thoracic- and Vascular Surgery, Philipps-University Marburg, 35043 Marburg, Germany; bartsch@med.uni-marburg.de (D.K.B.); gercken@staff.uni-marburg.de (N.K.-G.); 2Department of Gastroenterology and Endocrinology, Philipps-University Marburg, 35043 Marburg, Germany; sprengea@uni-marburg.de; 3Department of Diagnostic and Interventional Radiology, Philipps-University Marburg, 35043 Marburg, Germany; andreas.mahnken@staff.uni-marburg.de; 4Institute of Pathology, Philipps-University Marburg, 35043 Marburg, Germany; moritz.jesinghaus@uni-marburg.de; 5Department of Nuclear Medicine, Philipps-University Marburg, 35043 Marburg, Germany; friederike.eilsberger@uk-gm.de

**Keywords:** small intestine neuroendocrine tumor, locally advanced, unresectable, debulking, classification

## Abstract

In about 10% of patients with locally advanced, small intestine neuroendocrine neoplasm, a complete resection is not possible due to regional lymph node metastases with vascular involvement and the associated mesenteric desmoplasia. Debulking of the mesenteric mass in these symptomatic patients relieves symptoms in most patients. An accurate preoperative classification of the extent of the mesenteric mass and a vessel- and bowel-sparing surgical approach are important.

## 1. Introduction

Complete resection of the primary SI-NEN and its associated locoregional mesenteric disease, including nodal metastases and mesenteric desmoplasia, can be achieved in up to 90% of patients in experienced centers [1,2,3,4,5]. However, in 10% of the remaining patients the disease is locally advanced, and thus a complete resection (R0) is not possible. In SI-NENs, resectabilty is not determined by the primary tumor but by the extent of the mesenteric disease, regional lymph node metastases, and associated mesenteric desmoplasia along the branches of the superior mesenteric artery (SMA) to lymph nodes located adjacent to the root of the SMA and thence to para-aortic lymph nodes [6]. Several classifications have been proposed based on the relationship between mesenteric lymph node metastases and the superior mesenteric vessels, particularly with the goal of predicting potential complete resection [6,7,8,9]. Lymphadenectomy becomes more challenging in the presence of extensive mesenteric desmoplasia and mesenteric lymph node metastases towards the root of the superior mesenteric vessels [6]. ENETS recommendations [6] previously classified locally advanced or unresectable disease when tumor deposits and desmoplasia involved the SMA and the superior mesenteric vein (SMV). The central location of tumor deposits at the origin of the mesenteric root often causes symptoms related to bowel ischemia, kinking, and luminal obstruction [10,11,12,13]. In SI-NEN patients with locally advanced disease, cytoreductive or debulking surgery might be performed [3,4,5,14]. However, partial resection of the mesenteric disease, leaving behind significant disease centrally around the mesenteric root, may result in the loss of a significant length of the small bowel, which is associated with a high likelihood of persistent significant bowel symptoms. In addition, the also-persistent vascular comprise of the remaining small bowel might prohibit healing of a bowel anastomosis. However, it has been postulated that symptomatic relief can be achieved in more than 70% of patients, if the vessels of the mesenteric root can be decompressed [3,15]. Nevertheless, any attempt at partial resection of mesenteric disease should be critically discussed and indicated in a multidisciplinary context, with special consideration of other options for symptom control. To date, there is a paucity of outcome data regarding mesenteric debulking of well-defined, locally advanced SI-NENs. Therefore, the present retrospective study reports the outcome of mesenteric debulking of locally advanced, not completely (R0) resectable SI-NENs.

## 2. Methods

Patients who underwent surgery for SI-NENs between April 2012 and October 2024 were identified from the prospective database of the ENETS Center of Excellence Marburg University Hospital. Operated-on patients with a histopathological confirmed diagnosis were included, if:(a)Mesenteric disease, lymph node metastases, and mesenteric desmoplasia extended the level of the horizontal part of the duodenum (level 3 of the Marburg classification [9] (Figure 1), as demonstrated by functional and cross-sectional imaging and the operation report;(b)The tumor was classified as locally advanced or unresectable because mesenteric disease encircled the mesenteric vascular root above the level of the horizontal part of the duodenum and/or extended to the retroperitoneum (Table 1);(c)Patients were symptomatic with episodes of abdominal pain and/or bowel obstruction.

Patient characteristics, including age, sex, date of diagnosis, abdominal symptoms (abdominal pain, symptoms of intestinal obstruction), weight loss > 5 kg/2 months, the presence of carcinoid syndrome and carcinoid heart disease, previous therapies, previous surgical procedures, preoperative stage of disease, preoperative imaging, operative notes, histopathology, postoperative complications, and outcome were retrospectively analyzed.

Each SI-NEN patient was discussed in the interdisciplinary tumor board, and palliative debulking surgery of the mesenteric mass, including primary tumor resection, was offered to symptomatic patients without severe comorbidities.

Cross-sectional imaging with contrast-enhanced computed tomography (CT) and/or magnetic resonance imaging (MRI) and functional imaging with Ga68-DOTATOC PET/CT were re-reviewed by an experienced radiologist (A.M.) and nuclear medicine physician (F.E.). Imaging was re-analyzed regarding the presence of a pathological mesenteric mass >10 mm size above the level of the inferior wall of pancreatic body. Since there is no way to distinguish radiologically between an enlarged mesenteric lymph node and fibrotic mesenteric tissue, both were classified as a mesenteric mass. In addition, occlusion of the SMA, portal vein (PV), or SMV; retroperitoneal and/or pararaortal extension; advanced mesenteric desmoplasia, defined as ≥10 thick radiating strands within the mesentery [16]; and associated small bowel thickening > 3 mm were evaluated.

Operating reports were reviewed by two surgeons (D.K.B., E.M.) for extension of mesenteric disease according to the above-mentioned classification, procedures were performed, and the presence of advanced mesenteric desmoplasia was determined. Intraoperatively, mesenteric desmoplasia was defined as a white and stringy induration with shrinkage of the mesentery resulting in angulation of the small bowel that may have led to fixation of the mesenteric root to the retroperitoneum [17] (Figure 2).

Surgery was performed as previously described, with a vessel-sparing approach whenever technically possible [9,14]. In brief, all patients were operated on through an extended abdominal midline incision. If the primary tumor was not resected, the bowel was explored from the ligament of Treitz to the ileocecal valve to identify the primary lesion, possible multiple SI-NENs, and mesenteric lymph node metastases. The right colon and small bowel mesentery were always mobilized from posterior adhesions towards the retroperitoneum up to the level of the horizontal part of the duodenum and the lower border of the pancreatic body. Tumorous or fibrotic adhesions to the serosa of the horizontal duodenum frequently required sharp transection. With the mobilization, the mesenteric mass could be pulled down and its exact location on the mesenteric axis became clearer, facilitating anterior and posterior manipulation and dissection. In some cases, a full Cattell–Braasch maneuver [18] was required to make an accurate assessment and establish a strategy for resection. When this was not possible due to bulky mesenteric disease, intraoperative ultrasound was used to determine the extension of tumor deposits around the SMA and PV/SMV at the mesenteric root. In this series of locally advanced mesenteric disease (Table 1), the SMA and SMV were exposed by incision of the desmoplastic soft tissue or the metastatic lymph node tissue and were dissected ventrally free from tumor deposits in the sense of ventral decompression between the inferior border of the pancreatic body and the level of the inferior border of the horizontal part of the duodenum. This decompression should preserve the first two jejunal arteries and mesenteric venous branches, which are usually located in this region [19,20,21,22] (Figure 1). Preservation of these first two branches always resulted in an R2 resection, but this was accepted to preserve at least 200 cm of well-vascularized small bowel [23]. This was followed by retrograde resection towards the ileocecal vessels [9,14]. An attempt was always made to preserve the ileocolic vessels and important vascular collaterals and arcades along the bowel. In most cases, this was not technically possible, and mesenteric vessels were divided between the level of the horizontal part of the duodenum and the ileocolic branches (level 2). Dissection of the tumor and desmoplastic soft tissue along the SMA and SMV was performed using either bipolar forceps or scissors, or vessel-sealing devices. Resection of the involved bowel segment was delayed until dissection of the mesenteric tumor tissue was complete. When the ileocecal vessels had to be divided centrally, a right hemicolectomy was performed. Bowel continuity was reconstructed in all cases with a sutured, double-layer side-to-side anastomosis. In cases of macroscopically questionable perfused small bowel ends, indocyanine green near-infrared angiography was used to determine the blood supply at the anastomotic site [9,24].

Histopathological parameters analyzed included size of primary tumor(s), number of resected and metastatic lymph nodes, tumor grading assessed by Ki-67 proliferation index, and stage [25].

All patients received intravenous somatostatin (3 mg/24 h) 72 h perioperatively.

Postoperative complications were classified according to Dindo-Clavien [26]. Only clinically relevant complications ≥3 were considered for analysis. Reoperations or other local re-interventions due to bowel problems such as obstruction and/or ischemic perforation were recorded.

Patients were followed up with at the Departments Surgery or Gastroenterology of the Marburg University hospital until death or 31 December 2024. Follow-up examinations included an interview regarding the presence of abdominal pain and obstructive bowel symptoms such as constipation and vomiting, a physical examination, and cross-sectional imaging with MRI and/or functional somatostatin receptor imaging. The symptom of diarrhea was not analyzed, because it is often also caused by the presence of carcinoid syndrome. Follow-up examinations took place every 6 months according to ENETS guidelines [27,28].

### 2.1. Statistics

Variables were presented as frequencies with percentages for categorical variables and median with range for continuous variables as medians (interquartile ranges [IQRs]). Survival was defined as the time from the date of reintervention to death from disease or any cause of death and was determined by the Kaplan–Meier method. Statistical significance was defined as *p* < 0.05 for all analyses. Data analyses were performed with R and R Studio (v.1.1.456; RStudio, Inc., Boston, MA, USA).

### 2.2. Results

Over a 12-year period, 29 of 202 (14%) SI-NEN patients (23 male, 6 female) operated on, with a median age of 63 (46–78) years, had symptomatic, locally advanced disease. All 29 patients had abdominal symptoms, most commonly pain (*n* = 27, 93%) or episodes of bowel obstruction (*n* = 11, 38%). Twelve (41%) patients had a carcinoid syndrome, and one (3.5%) patient had carcinoid heart disease. Twenty-seven (93%) patients had stage IV disease and two (7%) stage III disease. Of the patients with stage IV disease, 23 (85%) had liver metastases, 20 (74%) had peritoneal metastases, and 4 (15%) had bone metastases. Prior to debulking surgery, 22 of 29 patients (76%) were treated with somatostatin analogues and 7 (24%) patients received PRRT. One patient had an angiographic stenting attempt in an occluded SMV. Fourteen (48%) patients had at least one previous SI-NEN procedure between 3 and 276 months prior to debulking surgery. Of these patients, 10 underwent initial resection of the primary tumor and 4 had only biopsies and/or bowel bypass procedures (Table 2).

All patients underwent preoperative cross-sectional imaging with CT and/or MRI as well as functional somatostatin receptor PET/CT imaging. Imaging revealed distant metastases in 26 of 29 (89%) patients, advanced mesenteric disease extending the level of the horizontal part of the duodenum in 15 (52%) patients (Figure 1), and tumor-associated occlusion or significant compression of the PV/SMV in 18 (62%) patients. Advanced mesenteric desmoplasia was postulated in 21 (72%) patients, and associated small bowel thickening wall was seen in 11 (38%) patients. The clinical characteristics are summarized in Table 2.

In 27 patients, surgery was planned, and in 2 patients, surgery was performed in an emergency setting due to massive gastrointestinal bleeding in one patient and acute bowel ischemia in the second patient. Debulking of the mesenteric mass with small bowel resection (*n* = 14, 48%), right hemicolectomy or ileocecal resection (*n* = 10, 38%), or resection of an ileotransversostomy (*n* = 2) was performed in 26 patients (Table 3). In one of these patients, the operation had to be performed in two steps due to massive intraoperative bleeding during the initial procedure, which ended with abdominal packing. The patient was successfully depacked after 48 h and the small bowel resection was completed. Another 2 patients (7%) underwent debulking of the mesenteric mass only. In one of these patients, the debulking of the mesenteric root had to be terminated prematurely due to an intraoperative cardiac arrest with resuscitation. After successful recovery, the 72-year-old female patient refused to undergo completion surgery. In the second patient, massive hemorrhage due to portal hypertension caused by occlusion of the SMV precluded meticulous debulking. One patient underwent resection of the ischemic bowel without debulking the mesenteric mass.

Intraoperatively, all 29 patients showed advanced mesenteric desmoplasia with extensive stringy induration and shrinkage of the mesentery resulting in small bowel angulation (Figure 2). Surgically, it was easier to dissect the mesenteric mass, when it had a “smooth”, round appearance (Figure 3A) rather than a “scalloped”, more infiltrative appearance (Figure 3B). Additional resection of liver metastases was performed in 14 (48%) patients. Macroscopically, no patient achieved a local R0 rection of the mesenteric mass. The median operative time was 262 (range 156–411) minutes. Twenty-six (90%) patients had at least 200 cm of small bowel preserved after debulking, and three (10%) patients had less than 150 cm of small bowel preserved (Table 3).

Histopathologic analysis revealed a singular SI-NEN in 12 patients and multifocal tumors in 10 patients, while 7 patients had a primary tumor removed in previous surgeries. The grading was G1 in 19 (66%) patients, G2 in 9 (31%) patients, and G3 NEC in 1 (3%) patient. The 78-year-old male patient with a SI-NEC (Ki67 50%) was diagnosed with a NET G2 21 months prior to his initial right hemicolectomy. A median of 20 (range 7–40) lymph nodes were resected during the debulking procedure (Table 3).

Four patients (14%) experienced clinically relevant postoperative complications requiring reoperation for bleeding (*n* = 2), intestinal obstruction due to torsion (*n* = 1), and bowel ischemia (*n* = 1). A 64-year-old male patient with postoperative bowel ischemia died on postoperative day 41 after three reoperations.

All but two patients (93%) patients reported postoperative improvement in abdominal symptoms. A total of 20 of 22 (91%) patients with preoperative episodes of abdominal pain reported significant improvement, and all 11 patients with preoperative episodes of bowel obstruction stated resolution of symptoms. Symptom improvement lasted from 4 months up to 142 months.

Postoperatively, 28 (97%) patients received additional treatment, including somatostatin analogues (*n* = 28), PRRT (*n* = 9), and TACE of liver metastases (*n* = 6). One patient with a locally advanced stage III SI-NEN underwent re-debulking of para-aortic and mesenteric lymph nodes after 82 months. Another patient required a surgical creation of an ileostomy for bowel obstruction 48 months after debulking surgery. A third patient underwent endoscopic duodenal stent placement for tumor-associated duodenal obstruction 5 months after debulking surgery.

After a median follow-up of 28 (1–142) months, 21 (72%) patients were alive with disease, 7 patients had died of the disease, and 1 patient had died of advanced renal insufficiency. The calculated 5-year survival rate was 70% (95%CI 0.5–0.89) (Table 3).

## 3. Discussion

Few studies have reported the outcome of surgery in the subgroup of locally advanced SI-NENs [5,9,14,29]. The clinical parameters of patients in these studies, including age, gender, symptoms, and disease stage, are similar to those in the present study. The present study, however, is the first to focus on the outcome of surgery in well-defined locally advanced, unresectable SI-NENs. Based on our experience, we specified the ENETS classification of locally advanced SI-NENs in more detail [6]. We classified the SI-NEN as locally unresectable when the mesenteric disease, either lymph node metastases or mesenteric fibrosis, involved the mesenteric root above the level of the inferior wall of the horizontal part of the duodenum and/or extended to the retroperitoneum (Table 1). Many of these patients are symptomatic from SMV obstruction [30], mesenteric ischemia [30], bleeding, or incipient small bowel obstruction [3,5,29]. Treatment options are limited. Management of these patients is complex and may require input from several disciplines, including gastroenterology, endocrinology, nuclear medicine, interventional radiology, surgery, and palliative care. In such cases, the use of aggressive surgery with resection of mesenteric vessels and their reconstruction has been reported [5,9,14,29]. It must be considered that the majority of patients have advanced, incurable stage IV disease with liver, peritoneal, and bone metastases and carcinoid syndrome, as in 85%, 74%, 15%, and 41% of patients in the present study. Therefore, the aggressiveness and extent of surgery has to be carefully balanced against the potential benefit of symptoms. Alternative options, e.g., venous and arterial stenting of the SMV and SMA for symptoms related to chronic ischemia, must be considered, as they have been shown to relieve the symptoms of obstruction in some patients [31,32]. We indicated surgery only in patients with bowel obstruction, bleeding, or clinical symptoms without significant comorbidities. The primary goal of surgery was always to perform a bowel-sparing resection of present primary tumors, as in 20 cases of the present study, and to decompress the mesenteric root by debulking the tumorous and desmoplastic tissue. Decompression was always started ventrally, and in some cases also extended dorsally, with great effort to keep the first two jejunal arteries intact to preserve at least 200 cm of perfused small bowel, which is the minimum length to avoid short bowel syndrome, according to guideline recommendations [2,6,10,33]. A recent well-designed, experimental human cadaver study supports this recommendation [22]. It demonstrated that the first two jejunal arteries originated from the left side of the SMA in two-thirds and from the posterior wall of the SMA in one-third of subjects, and the median small bowel length perfused by the first two jejunal arteries was 221 cm (85–238) [22]. The mean distance of origin of the first two jejunal arteries was 4.6 cm, and 6.0 cm. Another anatomical study has shown that in most patients, the first two jejunal arteries originate from the SMA to the level of the lower wall of the horizontal part of the duodenum [34]. In the present study, more than 200 cm of small bowel could be preserved in 24 (83%) patients by the described decompression procedure.

In 2011, Kitchens et al. described for the first time a partial abdominal evisceration with intestinal autotransplantation for complete resection of a locally advanced stage III SI-NEN with associated SMV occlusion in a 60-year-old male patient [35]. This patient was disease-free 28 months after surgery. This highly sophisticated procedure, prone to potential complications, is only indicated in the very rare situation of locally advanced SI-NEN without distant metastases. This could have been applied in two of the patients of the present study. These two patients were still alive with disease after primary tumor resection and debulking of the mesenteric disease for 85 and 142 months.

Decompression or debulking of the mesenteric root with or without bowel resection was a technically challenging procedure, as reflected by the median operative time of 262 (range 156–411) minutes. In three patients, two of them with SMV occlusion, mesenteric debulking had to be interrupted due to severe intraoperative bleeding. Overall, clinically relevant postoperative complications (CD ≥ 3) after debulking procedures occurred in 14% and 3.5% of the present study, which is comparable to the 22% and 0% reported in a Canadian study [34], but higher than the rates of 7% and 1% reported in a meta-analysis for the entire SI-NEN surgery [36].

In a Dutch series [37] and in a French series, 6% and 5% of patients [38], respectively, were unable to undergo any form of debulking surgery due to the presence of advanced peritoneal metastases, which resulted in a frozen abdomen that prevented surgical access. In the present study, we did not encounter this situation, maybe due to patient selection based on imaging.

In a previous study, the mesenteric mass could not be completely resected in 8 (14%) of 59 symptomatic SI-NEN cases and was instead debulked. Another three patients (5%) had mesenteric mass lesions that could not be resected or debulked and underwent an intestinal bypass [3]. Complete relief of abdominal obstructive symptoms was noted in all patients who underwent resection and/or debulking of the mesenteric nodal disease. In a Dutch study, 115 of 288 (40%) patients with stage IV disease and mesenteric fibrosis on imaging underwent surgery, including 61 (53%) patients with palliative intent for symptom control [37]. In this study, the extent of mesenteric disease, and thus local irresectability, was not defined, but resection or debulking of the mesenteric mass was performed in 50 (17%) patients. Results regarding symptom improvement were not reported, but the survival of patients with complete resection compared to those with residual mesenteric disease was not different [37]. In the present study, the mesenteric disease could be significantly debulked in 26 (90%) patients, and postoperative improvement in or disappearance of obstruction and pain was reported in over 90% patients. The symptom improvement was quite variable from 4 to 120 months.

A Canadian study of 59 SI-NEN patients reported a 5-year survival rate of 74% after aggressive debulking surgery [3]. In the present study, the actual 5-year survival rate was 70%. A Dutch study [37] found in a multivariate analysis that the 82-month median survival of SI-NEN patients who underwent resection of mesenteric mass was not significantly different from 100 months in patients who did not undergo resection (*p* = 0.485). In this study, no subgroup analysis of symptomatic patients with locally advanced disease was performed. Since patients with locally advanced SI-NENs in this and all other studies also received a variety of other treatments, including somatostatin analogues, PRRT, TACE, and everolimus, the effect of debulking surgery on survival is difficult to define unless an acute life-threatening situation such as bleeding or acute bowel obstruction was resolved. Thus, it remains open whether debulking surgery of mesenteric disease in symptomatic, locally advanced cases really prolongs survival.

Since most patients with locally advanced SI-NENs, 97% in the present study, receive somatostatin analogues, which can cause gallbladder complications, a preventive cholecystectomy should be considered during debulking surgery [39].

The present study has several limitations. These include its retrospective design, the lack of a control group of patients without mesenteric disease debulking, and the lack of a standardized, validated questionnaire to document abdominal symptoms. Its strength is the focus on patients with well-defined, locally advanced SI-NENs documented in a prospective database of a monitored ENETS Center of Excellence.

## 4. Conclusions

In conclusion, the present study provides evidence that debulking of the mesenteric mass in locally advanced, symptomatic SI-NENs is a technically demanding procedure from which the majority of patients benefit in terms of ischemic and obstructive bowel symptoms, if no major perioperative complications occur.

## Figures and Tables

**Figure 1 cancers-17-01318-f001:**
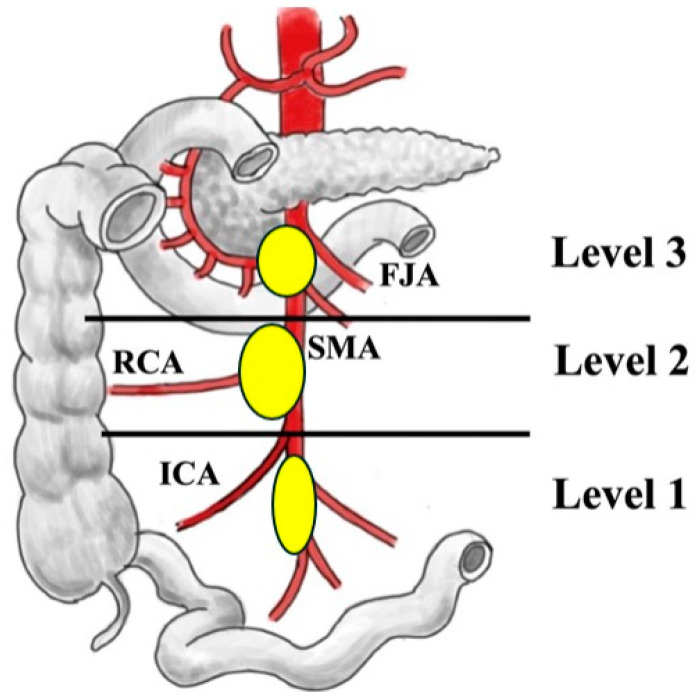
Classification of lymph node involvement [9]. FJA—first jejunal arteries; RCA—right colonic artery; SMA—superior mesenteric artery; ICA—ileocolic artery.

**Figure 2 cancers-17-01318-f002:**
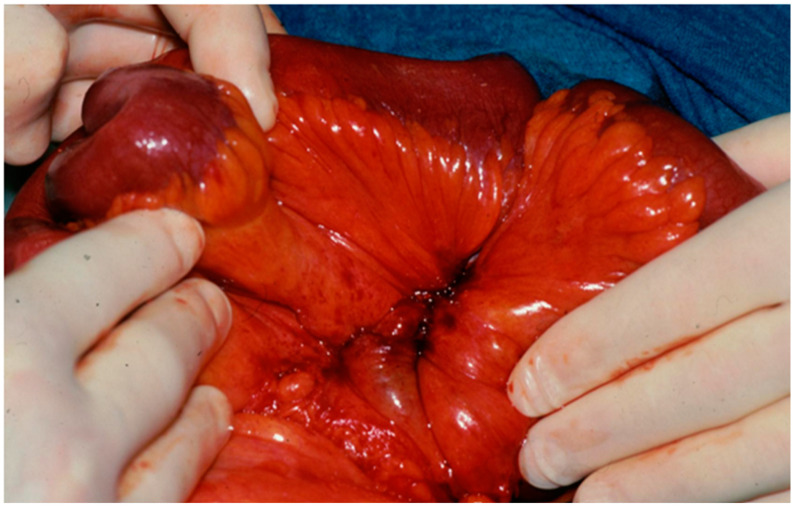
Mesenteric desmoplasia with shrinkage of the mesentery resulting in angulation of the small bowel that causes chronic ischemia (Department of Visceral, Thoracic and Vascular Surgery, Philipps University Marburg).

**Figure 3 cancers-17-01318-f003:**
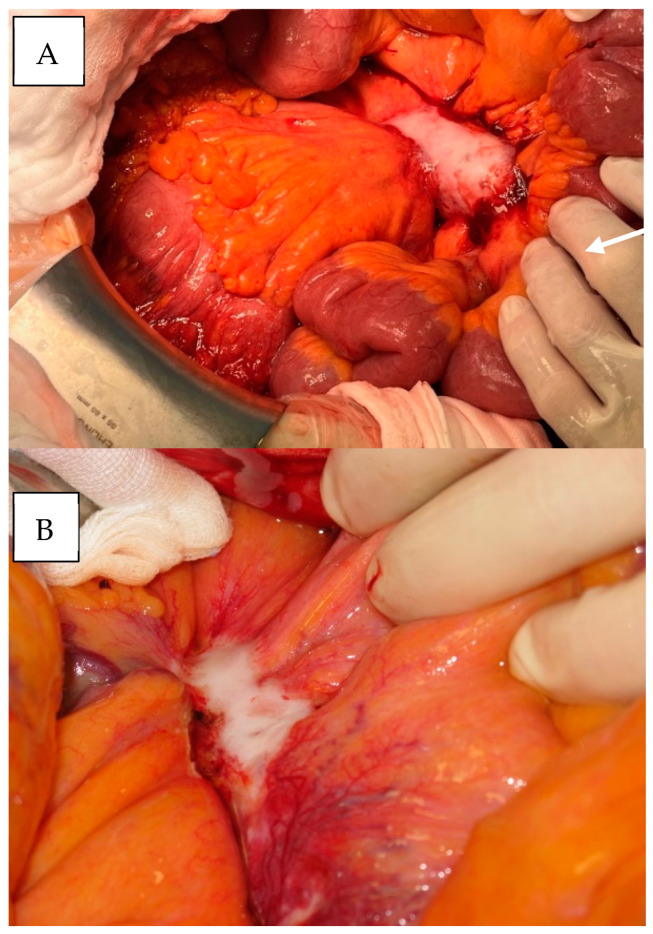
Smooth-like (**A**) and scalloped-like (**B**) mesenteric tumor appearance (Department of Visceral, Thoracic and Vascular Surgery, Philipps University Marburg).

**Table 1 cancers-17-01318-t001:** Proposed classification of local resectability in SI-NENs (modified according to 6).

Resectable	Mesenteric disease (nodal metastases and desmoplasia) up to the of the outlet of the ileocolic artery from the mesenteric superior artery
Borderline resectable	Mesenteric disease up to the level of the inferior pancreas body without encircling the mesenteric vessel root, nor the first two jejunal arteries, and without extension to the retroperitoneum
Locally advanced or unresectable	Mesenteric disease that encircles the mesenteric vessel root, including the first two jejunal arteries, above the level of the inferior wall of the horizontal part of the duodenum and/or extension to the retroperitoneum

**Table 2 cancers-17-01318-t002:** Clinical characteristics of locally advanced SI-NENs at the time of debulking surgery.

Proportion of locally advanced cases of all SI-NENs (2012–2024)	29/202 (14%)
Age, years median (range) at Dx	63 (46–78)
Gender (male/female)	23/6
Abdominal symptoms	29/29 (100%)
Abdominal pain	22/29 (76%)
Symptoms of bowel obstruction	11/29 (38%)
Weight loss	22/29 (76%)
Patients with stage III/stage IV disease	2 (7%)/27 (93%)
Previous SI-NEN surgery - With PT resection - Without PT resection (e.g., laparoscopic biopsy, jejunostomy, ileostomy)	14/29 (48%) 10/14 (71%) 4/14 (29%)
Occlusion or compression of PV/SMV on imaging	18/29 (62%)
Mesenteric mass > 10 mm at level 3 on imaging	16/29 (55%)
Advanced mesenteric desmoplasia on imaging *	21/29 (72%)
Previous SSA treatment	22/29 (76%)
Previous PRRT treatment	7/29 (24%)
Carcinoid syndrome	12/29 (41%)
Hedinger syndrome	1/29 (3%)
Distant metastases - Liver metastases Metastatic load >10% - Peritoneal metastases - Bone metastases	27/29 (93%) 23/27 (85%) 11/27 (41%) 20/27 (74%) 4/27 (15%)

Dx—diagnosis; PT—primary tumor; SMV—superior mesenteric vein; SSA—somatostatin analogues; PRRT—peptide radio receptor therapy, PV—portal vein, *—defined as ≥10 thick radiating strands within the mesentery [16].

**Table 3 cancers-17-01318-t003:** Surgery and pathological results and outcomes of patients with debulking surgery of locally advanced SI-NENs (*n* = 29).

Type of procedure - Debulking MM with small bowel resection - Debulking MM with right hemicolectomy/ileocecal resection - Debulking MM with resection of ileotransversostomy - Debulking MM without bowel resection - No debulking MM, but resection of ischemic bowel - Additional resection of liver metastases	Number *n* = 29 14/29 (48%) 10/29 (34%) 2/29 (7%) 2/29 (7%) 1/29 (3%) 14/29 (48%)
Operating time (min.), median (range)	262 (156–411)
Patients with advanced mesenteric desmoplasia intraoperatively *	29/29 (100%)
Patients with ≥200 cm small bowel length after debulking	26/29 (90%)
Grading, G1/G2/G3	19 (66%)/9 (31%)/1 (3%)
LN resected in patients with debulking, median (range)	20 (7–40)
Postoperative complications ≥CD 3	4/29 (14%)
Perioperative mortality	1/29 (3%)
Improvement in bowel obstruction	11/11 (100%)
Improvement in abdominal pain	20/22 (91%)
Additional SSA treatment	28/29 (97%)
Additional PRRT	9/29 (31%)
Additional local treatments (e.g., TACE, stenting SMA/SMV, duodenal stent)	9/29 (31%)
Additional abdominal surgery for SI-NENs	2/29 (7%)
Median follow-up (range) after debulking surgery, months	28 (1–142)
Patients alive with disease at study endpoint	21/29 (72%)
5-year survival after debulking surgery	0.7 (95%CI 0.5–0.89)

***** Defined as a stringy induration with shrinkage of the mesentery resulting in angulation of the small bowel. MM—mesenteric mass; CD—Clavien–Dindo [25]; SSA—somatostatin analogues; PRRT—peptide radio receptor therapy; TACE—trans-arterial chemo-embolization; SMA—superior mesenteric artery; SMV—superior mesenteric vein.

## Data Availability

The data presented in this study are available on request from the corresponding author.

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
