# Peer review of "Outcome of Debulking the Mesenteric Mass in Symptomatic Patients with Locally Advanced Small Intestine Neuroendocrine Tumors"

_cancers, 2025, doi:10.3390/cancers17081318_

Round 1
Reviewer 1 Report
Comments and Suggestions for Authors
The article is an interesting work on a subject with scarce scientific evidence, so it is necessary.
However, there are a couple of issues that need to be clarified.
On the one hand, the writing is very poor, with spelling errors, lack of spacing between characters, duplicated words... requires a detailed review and correction of these aspects.
On the other hand, there is a certain degree of confusion, since debulking surgery is mentioned throughout, but several patients achieved a radical R0 resection, so I would not apply this term.
Author Response
The article is an interesting work on a subject with scarce scientific evidence, so it is necessary. However, there are a couple of issues that need to be clarified.
On the one hand, the writing is very poor, with spelling errors, lack of spacing between characters, duplicated words... requires a detailed review and correction of these aspects.
Response: We apologize for the spelling errors. We thoroughly overworked the manuscript and corrected all typing errors and lacking spacing between characters.
On the other hand, there is a certain degree of confusion, since debulking surgery is mentioned throughout, but several patients achieved a radical R0 resection, so I would not apply this term.
Response: We agree that the notion in table 3 that 4 patients had an R0 resection according to pathology is somehow misleading, because none of the patients had a macroscopic local R0 resection as determined by the operating surgeon. Thus, all operations were indeed local debulking procedures. To avoid confusion, we deleted the sentence in lines 229/230 and the corresponding row in table 3 and stated in line 238 that macroscopically no patient achieved overall R0 resection.
Reviewer 2 Report
Comments and Suggestions for Authors
- Please include general patient data in the abstract's results section.
- Please correct manuscript by inserting a pause before all brackets.
- Was Figure 1 entirely made by you? If not, please indicate the reference.
- All clinicopathologic parameters that are listed in the results are not listed in the methods. Please correct, and keep the order you used in the methods. Same applies to Table 2.
- There is no mention at all about the histopathological subtype of these neuroendocrine tumors. Were they NETs or NECs?
- In Table 3, use capitals for each listed parameter.
- By mesenteric fibrosis do you mean the desmoplastic stromal reaction surrounding the tumor? Because based on your description, it is. It would be advisory to consult a pathologist regarding the matter, to be more accurate.
- It would be advisory to include pathological stage, as well.
- Your results are not listed in the discussion section.
Author Response
- Please include general patient data in the abstract's results section.
Response: The reviewer is right; the abstract could contain more information about the patient than their age. Due to the 250-word limit, we had to omit this in favour of other information. As suggested, however, we now added gender and the presence of abdominal symptoms.
- Please correct manuscript by inserting a pause before all brackets.
Response: We apologize and corrected the manuscript accordingly.
- Was Figure 1 entirely made by you? If not, please indicate the reference.
Response: Indeed, the illustration was created by us. A similar figure was published previously by our group (Bartsch et al., Cancers 2022, refence no. 9) as already indicated in the figure legend.
- All clinicopathologic parameters that are listed in the results are not listed in the methods. Please correct, and keep the order you used in the methods. Same applies to Table 2.
Response: We agree and have supplemented the methods section regarding the clinicopathological parameters (lines 88-90 and 154-156).
- There is no mention at all about the histopathological subtype of these neuroendocrine tumors. Were they NETs or NECs?
Response: We already mentioned the the grading in the result section as follows (lines 247-249): “The grading was G1 in 19(66%) patients, G2 in 9 (31%) patients and G3 NEC in one (3%) patient, respectively. The 78-year-male patient with a SI-NEC (Ki67 50%) was diagnosed with a NET G2 21 months prior to his initial right hemicolectomy.” In addition, it was listed in table 3.
- In Table 3, use capitals for each listed parameter.
Response: We corrected the table as suggested.
- By mesenteric fibrosis do you mean the desmoplastic stromal reaction surrounding the tumor? Because based on your description, it is. It would be advisory to consult a pathologist regarding the matter, to be more accurate.
Response: The terms mesenteric fibrosis and mesenteric desmoplasia are very closely related and sometimes used synonymously. The reviewer is correct that we meant mesenteric desmoplasia. Mesenteric desmoplasia specifically describes fibrosis triggered by a tumor, most commonly by a SI-NET. Thus, the reviewer is correct that we meant mesenteric desmoplasia, that leads to mesenteric retraction, vascular encasement and/or ischemia due to compression of blood vessels as defined defined in the method section (lines 107-110). This is a clinical definition, which does not need a detailed pathological analysis. To be more accurate, we now changed the term to mesenteric desmoplasia throughout the text (lines 103, 108, 200, 232 tables 2). We have added another figure (new Figure 2) for illustration purposes.
- It would be advisory to include pathological stage, as well.
Response: As suggest the tumor stage was now described in the result section (lines 186-187) as follows: “27 (93%) patients had stage IV disease, and 2 (7%) patients had stage III disease. It was already mentioned in table 2.
- Your results are not listed in the discussion section.
Response: We not completely agree with the reviewer. We already discussed our results of debulking, including e.g. advanced stage IV (lines 277-280), indication for surgery (lines 282-286), preservation of small bowel length (lines 297-299), complications (lines 310-312), improvement of symptoms (lines 330-333) and survival (lines 340-343), with the existing literature. In addition, we now added as suggested the following information as follows (lines 273-274): “The clinical parameters of patients in these studies, including age, gender, symptoms, disease stage, are similar to those in the present study.”
Reviewer 3 Report
Comments and Suggestions for Authors
Papers that deal with neuroendocrine neoplasms with extreme precision as in the case of colleagues who present this study are not very frequent. Unfortunately, neuroendocrine neoplasms are increasing in the population and if up to 15 years ago the age of onset was in the sixth seventh decade of life, now we can also find them in pediatric age (doi.org/10.3390/cancers16203440 to be read and cited in the bibliography). The diagnosis of neuroendocrine neoplasia is generally made during tests conducted for another pathology or for the complications that patients with such pathologies encounter. All organs can be affected and in 30% of cases found in the ileum they are multifocal. Our colleagues have correctly studied the cases, not a few, that have come to their attention. The multidisciplinary discussion to which they have submitted the individual cases was excellent. The attempted debulking as a rescue maneuver for intestinal hypovascularization well described, we ask, if they have them, the photos of what they define as mesenteric fibrosis and that we suspect is a neoplastic lymphangitis. We absolutely agree in introducing somatostatin in therapy with attention to the complications of this drug (PMID: 38051513 to be cited for completeness in the bibliography), which perhaps recommend a preventive cholecystectomy. We also find it right to introduce everolimus in therapy and let's not forget that immunotherapy in cases where it is possible to use them gives good results. One last thing, we recommend introducing a table in which there are the cases G1, G2, G3. to make the paper even clearer. Excellent and praiseworthy the citation on the emergence of the first two jejunal arteries to the right of the mesenteric. iconography to be improved, excellent English, good bibliography
Author Response
Papers that deal with neuroendocrine neoplasms with extreme precision as in the case of colleagues who present this study are not very frequent. Unfortunately, neuroendocrine neoplasms are increasing in the population and if up to 15 years ago the age of onset was in the sixth seventh decade of life, now we can also find them in pediatric age (doi.org/10.3390/cancers16203440 to be read and cited in the bibliography).
Response: We absolutely agree with reviewer that GEP-NET nowadays can be also sometimes in children and adolescents. However, the proposed reference (Mastrangelo S et al., Cancers 2024 Oct 10;16(20):3440) deals exclusively with appendiceal NET, which is not the focus of our study. Therefore, we prefer not to include the reference in the bibliography.
The diagnosis of neuroendocrine neoplasia is generally made during tests conducted for another pathology or for the complications that patients with such pathologies encounter. All organs can be affected and in 30% of cases found in the ileum they are multifocal. Our colleagues have correctly studied the cases, not a few, that have come to their attention. The multidisciplinary discussion to which they have submitted the individual cases was excellent. The attempted debulking as a rescue maneuver for intestinal hypovascularization well described, we ask, if they have them, the photos of what they define as mesenteric fibrosis and that we suspect is a neoplastic lymphangitis. We absolutely agree in introducing somatostatin in therapy with attention to the complications of this drug (PMID: 38051513 to be cited for completeness in the bibliography), which perhaps recommend a preventive cholecystectomy.
Response: As suggested, we cited the proposed refence (Calomino N et al., Ann Ital Chir. 2023;94:518-522) now in the discussion section as new reference 38 and added the following statement (lines 356-358): Since the vast majority of patients with locally advanced SI-NEN, 97% in the present study, receive somatostatin analogues which can cause gallbladder complications, a preventive cholecystectomy should be considered during debulking surgery [38].
We already included 2 photographs of a mesenteric mass (old Figure 2A/2B). As suggested, we added an additional photograph (new Figure 2) demonstrating mesenteric desmoplasia as defined in the method section.
Figure 2: Mesenteric desmoplasia with shrinkage of the mesentery resulting in angulation of the small bowel that causes chronic ischemia.
We also find it right to introduce everolimus in therapy and let's not forget that grapg with immunotherapy in cases where it is possible to use them gives good results. One last thing, we recommend introducing a table in which there are the cases G1, G2, G3. to make the paper even clearer.
Response: We agree with the reviewer. The grading was already mentioned in the result section (lines 247-249) as follows: “The grading was G1 in 19(66%) patients, G2 in 9 (31%) patients and G3 NEC in one (3%) patient, respectively. The 78-year-male patient with a SI-NEC (Ki67 50%) was diagnosed with a NET G2 21 months prior to his initial right hemicolectomy.” In addition, the grading is also listed in table 3. We think that an additional table for grading of the patients does not give more relevant information.
Excellent and praiseworthy the citation on the emergence of the first two jejunal arteries to the right of the mesenteric. iconography to be improved, excellent English, good bibliography
Response: We thank the reviewer for the generous comment. As suggested, we overworked the manuscript and improved iconography.
Round 2
Reviewer 1 Report
Comments and Suggestions for Authors
The corrections performed have improved the quality of the paper.